# Ecotoxicological Assessment of Polluted Soils One Year after the Application of Different Soil Remediation Techniques

**DOI:** 10.3390/toxics11040298

**Published:** 2023-03-24

**Authors:** Mario Paniagua-López, Antonio Aguilar-Garrido, José Contero-Hurtado, Inmaculada García-Romera, Manuel Sierra-Aragón, Ana Romero-Freire

**Affiliations:** 1Departamento de Edafología y Química Agrícola, Faculty of Science, University of Granada, 18071 Granada, Spain; antonioag@ugr.es (A.A.-G.); pepecontero@ugr.es (J.C.-H.); msierra@ugr.es (M.S.-A.); 2Departamento de Microbiología del Suelo y Sistemas Simbióticos, Estación Experimental del Zaidín, Consejo Superior de Investigaciones Científicas (EEZ-CSIC), 18008 Granada, Spain; inmaculada.garcia@eez.csic.es

**Keywords:** soil pollution, circular economy, remediation, bioavailability, ecotoxicology, bioassay, *Hordeum vulgare*, *Lactuca sativa*, *Raphidocelis subcapitata*, *Daphnia magna*

## Abstract

The present work evaluated the influence of eight different soil remediation techniques, based on the use of residual materials (gypsum, marble, vermicompost) on the reduction in metal(loid)s toxicity (Cu, Zn, As, Pb and Cd) in a polluted natural area. Selected remediation treatments were applied in a field exposed to real conditions and they were evaluated one year after the application. More specifically, five ecotoxicological tests were carried out using different organisms on either the solid or the aqueous (leachate) fraction of the amended soils. Likewise, the main soil properties and the total, water-soluble and bioavailable metal fractions were determined to evaluate their influence on soil toxicity. According to the toxicity bioassays performed, the response of organisms to the treatments differed depending on whether the solid or the aqueous fraction was used. Our results highlighted that the use of a single bioassay may not be sufficient as an indicator of toxicity pathways to select soil remediation methods, so that the joint determination of metal availability and ecotoxicological response will be determinant for the correct establishment of any remediation technique carried out under natural conditions. Our results indicated that, of the different treatments used, the best technique for the remediation of metal(loid)s toxicity was the addition of marble sludge with vermicompost.

## 1. Introduction

Soil pollution by potentially toxic elements (PTEs) is one of the most pressing public concerns, since it could directly affect not only the environment but also public health. It denotes a major degradation of the ecosystem [1] and a significant potential toxicological risk to organisms [2]. Over 10 million sites worldwide were reported with soil pollution by several anthropogenic sources and activities (industry, mining, smelters, agriculture, etc.) [3], and, therefore, the correct management of these sites is essential to avoid damages to human health or the environment. Of the usual sources of soil pollution, the current pace and development of the mining industry is a major concern worldwide, mainly due to the pollution by heavy metal(loid)s (e.g., Cr, Hg, Ni, Cu, Zn, Cd, As, Pb). They can accumulate and persist in soils, affecting ecosystem functions, compromising soil quality and the balance of ecosystem communities [4,5].

The Aznalcóllar disaster (Seville), considered one of the largest mining disasters in Europe [6], is a notable example of this type of pollution and the damage that can be caused by PTEs. In this mining accident a total of 45 km^2^ of mainly agricultural soils were affected by the discharge of 45 × 105 m^3^ of acidic waters and toxic tailings after the rupture of the Aznalcóllar pyrite mine sludge pond in 1998 [7]. Although the environmental impact was greatly minimized, due to the high buffering capacity of the soils [8,9], one of the most ambitious soil remediation programs in Europe was carried out. However, more than 20 years after the accident, residual pollution is still found in the area [10], posing a potential toxic risk to living organisms [11,12]. In the first 18 km downstream of the tailings pond, unvegetated soil patches remained, with high concentrations of PTEs and unfavorable physicochemical properties (i.e., acidic pH, low organic matter content, etc.) [13]. These areas represent approximately 7% of the total area affected, and can act as a source of diffuse contamination that must be addressed [14]. According to [12], these unvegetated soil patches pose an environmental risk, assessed by risk quotients [15] for the total concentration of As, Cu and Pb, and for Cu and Zn when considering water-soluble concentrations.

Recently, the development of a large number of safer, cleaner, less expensive and more environmentally friendly soil remediation techniques was studied [16]. In the case of residual pollution, the usual techniques aimed to reduce the toxicity of pollutants in the soil by controlling their mobility and bioavailability, since the toxicity risk strongly depends on them, rather than of their total concentration [17,18,19]. Thus, the remediation of soils affected by residual pollution must consider the main soil properties related to the reduction in mobility, availability and toxicity such as pH, organic matter, calcium carbonate content, texture, cation exchange capacity and iron oxy(hydr)oxides [12,18,20,21]. One of the most widely applied remediation technologies is the addition of amendments to polluted soils, especially low-cost ones, such as waste from several human activities, to tackle both the negative impacts of PTEs on the environment and to connect with the circular economy strategy [22,23,24]. In this sense, some research explored the use of materials at the end-of-life-cycle from different sectors to remediate residually polluted soils by mining activity. For example, marble waste sludge amendment can significantly reduce soil acidity and PTEs mobility in polluted mines by shifting the water-soluble forms of PTEs to fractions associated with carbonates, metaloxides, organic matter and insoluble secondary oxidation minerals and residual phases [25]. Likewise, it was also demonstrated that the application of organic amendments on the non-vegetated soil patches in Aznalcóllar area resulted in significant changes in the main soil properties, as well as in reductions in the soluble and exchangeable forms of several PTEs [13]. However, caution should be exercised as, in some cases, the addition of amendments to residually polluted soils was reported to be ineffective in remediating some PTEs (e.g., As), in relation to changes in environmental conditions (reducing or oxidizing conditions) [26,27]. Nevertheless, further in situ techniques are available and in use. For example, landfarming (periodic tilling of the polluted soils to remove surface crusts and aerate) and biopiles (mixing polluted soils with recovered soils) were applied together with composting in [12] in bare patches of the Aznalcóllar area with the aim of reducing the mobility and bioavailability of soil pollutants.

In order to ensure that remediation techniques are effective, the determination of PTEs forms at present is not enough, and the potential effect that the studied PTEs can cause in living organisms is essential. Commonly, the assessment of the potential harmful effect of the PTEs in soils is carried out by obtaining direct responses of living organisms exposed to them. However, the effects of the pollutants on organisms will greatly differ due to different uptake pathways, thus requiring the use of a set of organisms for a more realistic ecotoxicity assessment [28]. Therefore, multiple toxicity tests must be conducted with representative species of different taxonomic groups to evaluate toxic effects. 

The aim of the present work is to assess if the use of different remediation techniques, based on the use of residual materials (circular economy), could be helpful in reducing the metal(loid)s toxicity in a polluted natural area. To perform a complete toxicity assessment, selected amendments were applied in field exposed to real conditions and they were evaluated one year after the application. More specifically, five ecotoxicological tests, on either the solid or the aqueous (leachate) fraction of the amended soils, were performed. The main soil properties were also measured and the total, water-soluble and bioavailable metal content (Cu, Zn, As, Pb and Cd) were determined to evaluate which properties and metal fraction most influence soil toxicity and to select the best remediation technique.

## 2. Materials and Methods

### 2.1. Study Area and Remediation Treatments

The study site was located in the area affected by the Aznalcóllar mine spill (Seville, SW Spain), in the sector near the mine and highly affected by the pyritic tailings spill in 1998. The high contamination soil levels were evidenced by the presence of soil patches of varied surface with no vegetation growth and heterogeneously dispersed among re-vegetated areas [14]. Within the affected area, in an unvegetated soil patch near the mine, different remediation treatments were tested (Figure 1).

Specifically, an experimental plot of 32 m^2^ consisting of 8 subplots (4 m^2^ each) with eight different treatments was established (Table 1). In all cases, the dose of amendment applied, both inorganic (gypsum and marble) and organic (vermicompost), was 5 kg m^−2^ (equivalent to 50 t ha^−1^). In addition, one vegetated subplot adjacent to the unvegetated soil patch to the experimental plot was selected as recovered control (RS), and one subplot within the unvegetated soil patch as unrecovered control (CS).

### 2.2. Soil Properties

After one year of exposure to the different remediation techniques used, soil samples, in each studied subplot, were collected by mixing 200 g of top soil (0–10 cm) from each corner and midpoint of a square 1 m per side (composite samples; n = 3). Samples were air dried at room temperature in laboratory and passed through a 2 mm sieve. This fraction was used to characterize the main soil properties and for the toxicity bioassays, and when determinations required, soil samples were also finely ground. Soil pH was determined in a soil/water ratio 1:2.5 with a 914 pH/Conductometer Metrohm (Herisau, Switzerland); soil/water extract (1:5) was prepared to determine the electrical conductivity (EC, dS/m) using a Eutech CON700 conductivity-meter (Oakton Instruments, Vernon-Hills, IL, USA); calcium carbonate content (%CO_3_Ca) by volumetric gases [29]; organic carbon (%OC) was determined according to [30]; total nitrogen (%N) was analyzed by dry combustion using an elemental analyzer TruSpec CN LECO^®^ (St. Joseph, MI, USA); soil texture was determined by the Robinson pipette method [31]; base content and cation exchange capacity (CEC) were determined according to the methodology of [31]; and the water field capacity (%WFC) and available water content (%AWC) were determined from Richards methodology [32].

### 2.3. Soil Metal Content Determination

To determine total metal content (xx_T), soil samples were acid-digested (HCl:HNO_3_, 3:1); water-soluble content (xx_W) was obtained from soil/water extract (1:5) [33], and bioavailable content (xx_B) was obtained by extraction with 0.05 M EDTA (pH 7), as described by [17]. All samples were analyzed by ICP-MS in a PerkinElmer NexION^®^ 300D spectrometer (Waltham, MA, USA). The ICP-MS operating conditions included three replicates in each measurement. For calibration, two sets of multi-element standards containing all the analytes of interest were prepared using rhodium as the internal standard. All standards were prepared from ICP single-element (Zn, Cu, Pb, As and Cd) standard solutions (Merck, Darmstadt, Germany) after dilution with 10% HNO_3_. The accuracy of the method was confirmed by analyzing Certified Reference Material CRM025-050 (Sandy Loam Soil), six replicates. For Zn, Cu, Pb, As and Cd, the average recoveries were 98.4 ± 12% of the CRM.

### 2.4. Ecotoxicological Approach

#### 2.4.1. Ecotoxicological Tests in Soil-Solid Fraction

Root elongation toxicity test with *Hordeum vulgare* L. (barley, monocotyledonous, Poaceae) evaluates the toxic effect directly in soil fraction [34]. Soil moisture content was fixed at WFC and soils were incubated in cylindrical pots (8 cm in diameter and 11 cm in height). Barley seeds were pregerminated at 20 °C in darkness 48 h in Petri dishes. Six seeds with a radical length lower than 2 mm were planted in each pot, approximately 10 mm beneath the surface of the soil. Plants were placed under controlled growing conditions (16 h/8 h light/darkness, 20 ± 2 °C). After 5 days of growth, each soil was gently washed out of the pots. Plant roots were carefully washed, preventing root damage and the result was expressed as the measured longest root per seedling (cm).

Heterotrophic soil respiration (Rs) was measured by determining the CO_2_ flux from studied soils with a Microbiological Analyser μ-Trac 4200 SY-LAB model (SY-LAB Geräte GmbH, Neupurkersdorf, Austria) [35]. Soil moisture content was fixed at WFC and soils were incubated at a constant temperature of 30 °C with addition of glucose (3 mg per gram of soil). CO_2_ production was determined by absorption during 6 h in vessels with two electrodes containing a solution of potassium hydroxide (KOH 0.2%) and hermetically sealed. The results obtained indicate the average of induced soil respiration and were expressed as the µg of CO_2_ respired for day and gram of soil (µg CO_2_ day^−1^ g^−1^ soil).

#### 2.4.2. Ecotoxicological Tests in Soil-Water Fraction

A root elongation toxicity test with *Lactuca sativa* L. (lettuce, dicotyledonous, Asteraceae) was performed according to a modification of US EPA recommendations [36]. In Petri dishes, 15 seeds of *L. sativa* were incubated in 5 mL of soil/water extract (1:5) from the studied soils. The dishes were placed in an incubator at 25 ± 1 °C and the length of the developing roots was measured after 120 h. The endpoint calculated was the % of root elongation reduction compared to a control performed only with distilled water.

The algal growth inhibition test with *Raphidocelis subcapitata* (freshwater algae) was carried out based on [37]. Exponentially growing algae (10^4^ cell mL^−1^) were exposed to soil/water extract from the studied soils over a period of 72 h under defined conditions, as described elsewhere [38]. Growth was quantified from measurements of the algal biomass density (cell counts) by spectrometry (670 nm) as a function of time. The specific growth rate of *R. subcapitata* was calculated from the logarithmic increase in cell density in the intervals from 0 to 72 h. The results were expressed as the mean of the % inhibition of the algae growth of the sample compared with a negative control (ISO media).

*Daphnia magna* (Crustacea) immobilization test was carried out according to OECD 202 guideline [39]. Daphnid neonates, up to 24 h old, were transferred to 6-well polycarbonate test plates with soil/water extract and incubated for 48 h under control conditions (darkness and 20 ± 2 °C) without feeding for the duration of the experiment. After 48 h of exposure, daphnids were observed under magnification and those that were not able to swim within 15 s under gentle agitation were recorded as immobilized. The calculated endpoint was the % of immobilized daphnids compared to the control performed with distilled water.

### 2.5. Statistical Analysis

Data distributions were established by calculating the mean values and the standard deviations by cumulative frequency–distribution curves. The differences between the individual means of the study samples were compared using ANOVA and Duncan post hoc test (*p* < 0.05) using SPSS v.21.0 (SPSS Inc. Chicago, IL, USA). In order to study the relation between soil properties, metal content and ecotoxicological approach in the studied soils, a non-metric multidimensional scaling (NMDS) analysis was carried out using the “vegan” package of RStudio software (RStudio Inc. (R), 250 Northern Ave, Boston, MA, USA).

## 3. Results and Discussion

### 3.1. Improvement of Soil Properties after Remediation

The use of the different techniques selected for the remediation of contaminated soil showed efficient results in the increase in soil pH (Table 2). The acidic solution from the oxidation of the pyrite-tailings infiltrated the soil and the H^+^ were neutralized by exchangeable bases, by weathering of silicate mineral or, more intensely, by carbonates [40]. According to our results, the use of marble (M) was the best treatment to reach pH near neutrality. This amendment was the one that contributed the highest CO_3_Ca values, this being the main factor affecting the increase in pH. According to [41], the potential acidity of 1 g of pyritic sulfur was neutralized by approximately 3 g of CO_3_Ca and, for this reason, liming was found to be one of the most widespread remediation actions to control soil acidity issues [42]. For all remediation amendments, the electric conductivity (EC), as a measurement of salinity, decreased by half, but without reaching the values present in the recovered soil (RS). The use of vermicompost, as part of the amendments, was directly related to the higher increase in OC and N content (BV, GV, LV and MV). The addition of compost increased the content of stable organic compounds in soils [43], improved chemical soil fertility of contaminated soil promoting favorable conditions for plants and microorganism [44] and reduced the mobility of potential toxic elements [45]. Vermicompost increased OC content in soils, but this content did not reach the values observed in RS, where natural vegetation appeared to improve soil properties related to fertility [46]. In general, the use of the selected amendments showed a slight increase in the CEC of soils, higher when vermicompost was used in the mixture. It was the landfarming + vermicompost (LV) and biopiles + vermicompost (BV) which showed CEC values close to those of the recovered soil (RS). This was consistent with [12], the authors of which reported a positive correlation between soil OC content and CEC in soils that remained contaminated 18 years after the Aznalcóllar accident and that were treated at laboratory scale with landfarming, biopiles and composting. The use of the selected amendments modified soil texture as well, with the higher values of fine granulometries (silt and clay) in BV. All applied amendments improved soil water retention. This may be related to the increased OC content in the amended soils, which increased soil water retention and infiltration capacity [47,48], improved soil structural stability [49] and reduced soil bulk density [50].

### 3.2. Availability and Solubility of the Studied Elements

Table 3 shows the total (Xx_T), bioavailable (Xx_B), and water-soluble (Xx_W) content of Cu, Zn, As, Pb and Cd in recovered soil (RS), contaminated soil (CS), and contaminated soil after the application of the amendments of study and incubation in field during one year.

Heavy metal pollution is usually determined based on the total metal content in soils, and there are several legislations that specify total concentration values that are intervention values for polluted soils. According to the Andalusian government, where soil samples are located, the intervention values for Cu, Zn, As, Pb and Cd are 595, 10,000, 36, 275 and 25 mg/kg, respectively [51]. According to this regulation, only total As and Pb showed contents above the guideline values. However, even nowadays, there is not a clear consensus for determining the metal guideline values in soils. Not all countries defined the corresponding guidelines, and when compared, the concentrations differed significantly from one place to another (See Appendix A for more detail). In Spain, intervention values for some elements vary more than 10 times depending on the regional government. This is the case of Galicia guideline (northwest Spain), where for Cu, Zn and Cd guidelines, the values are 50, 300 and 2 mg/kg, more than 10 times lower than the Andalusian values [52]. If we compare the total content of these three elements in our soils with the Galician legislation, we also have the risk of contamination for Cu, Zn and Cd that, in principle, were out of risk in Andalusia. In the case of other countries, values differed even more, i.e., in China, the risk screening value for Cd is 0.2 mg/kg [53], or in Canada, the intervention value for As is 12 mg/kg [54] (Appendix A). Thus, the risk of pollution could be over or under-estimated when studies are based in total concentrations only, and it is dependent on the local regulations and guideline values. Furthermore, it is not expected that the use of soil amendments modifies the total heavy metal content, as it was the case for the treatments used in this study, their influence being more related to the available metal forms. Indeed, the total concentration of a PTE may not be relevant in terms of toxicity and its behavior in the soil, and its ecotoxicity being more related to its mobility in soil [55]. Therefore, it is also important to determine the available forms of PTEs before and after remediation, and regulations be based on them and not only in total concentrations, since they are more accurately related to their actual risk of toxicity.

In the case of bioavailable content (Xx_B) obtained by extraction with EDTA, heavy metal concentrations were even higher in the recovered soil (SR) than in the contaminated one (CS). These results agreed with those reported by [13,56], in which an increase in EDTA-extractable Cu, As, Pb and Cd with increasing soil organic matter content was reported, as organic matter can be considered as a key soil property for the extractability of metals by chelators such as EDTA [57,58]. Additionally, the use of the selected amendments decreased for all studied elements their bioavailable content compared to the CS, the application of marble sludge (M) being the remediation technique that showed the highest decrease in bioavailability for all elements. Marble sludge influenced the increase in the CO_3_Ca content of studied soils the most, which implies that heavy metal could precipitate as insoluble carbonates [12,42] and be extracted by the EDTA [59].

Moreover, the determination of water-soluble forms is one of the easiest and cheapest techniques to study metal pollution. The water-soluble forms could also be a good indicator of metal pollution. In this regard, the water-soluble concentrations (Xx_W) observed were greater in the contaminated soils compared with the recovered one in all cases, except for As and Pb. Regarding As, authors such as [14,60] reported increases in As solubility in recovered soils in the Guadiamar Green Corridor due to an increase in the organic matter content of the soils over time, 15 and 20 years after the pyritic sludge spill, respectively. If we compare our results with those reported in [61], which showed the maximum contaminant level in mg/L for the different studied elements, we obtained that, in our case, after unit transformation, we obtained values higher than the maximum contaminant level for contaminated soil (CS) in the case of Cu (0.25 mg/L), Zn (0.8 mg/L) and Cd (0.01 mg/L). That is, water-soluble content decreased with the amendments, while after the application of biopiles (B and BV) and landfarming (L and LV) techniques, and according to the related values, water-soluble concentrations in these cases were still higher than the maximum contaminant levels established for Zn and Cd, which implies that they are not feasible treatments for controlling metal pollution.

### 3.3. Effects of Metal Toxicity in Different Soil Organisms after Remediation

The determination of only the metal content, both total and in different available forms, could give some idea of the degree of decontamination of a soil, but it is not enough to address a soil as not polluted. Therefore, the study of the behavior of living organisms in these soils by using toxicity bioassays is essential for a correct ecotoxicological risk assessment (ERA) [62]. Exposure to high concentrations of different PTEs poses a serious threat to organisms [63], as they can reduce microbial biomass [64], affect the taxonomic diversity of soil communities [65], and act on various soil microbial processes, thus disrupting nutrient cycling and the capability to perform key ecological functions (e.g., mineralization of organic compounds and synthesis of organic substances) [66,67].

Ref. [19] observed high variability when different bioassays were compared for studying soil toxicity. Therefore, selecting different living organisms is crucial to obtain real approaches of remediation results. In the present study, residual toxicity after remediation was evaluated by a total of five different well-established ecotoxicological tests using different organisms and soil fractions. In soil-solid fraction, toxicity was determined by barley and soil basal respiration, whereas in soil-water fraction lettuce, daphnid and algae were used to determine toxicity (Figure 2).

According to the tests performed in soil-solid fraction, we observed a positive response for all treatments compared to the contaminated soil (CS). In the toxicity test using barley, root elongation was close to that obtained in the recovered soil (RS) for all remediation techniques, with the highest root elongations in the case of addition of gypsum (G, GV) and marble (M, MV). In the case of soil respiration, it showed recoveries compared to the CS for all treatments. In general, for this bioassay, the addition of vermicompost showed higher respiration in all cases except in marble with vermicompost (MV), where values were similar to the application of marble alone. B and L techniques alone, without their combination with vermicompost, showed the lowest increases in soil respiration compared to CS.

In the toxicity tests performed in soil-water fraction, results obtained from the different treatments were not compared to the recovered soil, but to the response obtained from the control samples performed with distilled water instead of with soil extract. In relation to these tests, for the lettuce root reduction, total inhibition of root growth was found for CS, while no toxic effect was found under all remediation treatments. However, G and M stood out as the treatments that led to the most similar response to RS, enhancing the root growth to a greater extent. In the case of the test carried out by the algae, we observed no toxic effects for GV, M and MV and, in the case of other remediation techniques, the algae inhibition was even higher than in the CS in some cases (B, BV, G, L, LV). The test studying daphnid mortality showed results similar to the algae, and only the GV, M and MV presented a decrease in the toxicity for this organism, whereas the other remediation techniques used showed high toxicity, even with 100% of mortality in BV, L and LV, similar to the CS.

To assess environmentally relevant soil toxicity, the use of a diverse set of exposure routes is recommended; thus, the set of bioassays selected should jointly take into account exposure to both the solid and the soluble phases of the soil [68]. In this study, according to the toxicity bioassays performed, the response of the organisms to the treatments differed depending on whether the solid fraction or the aqueous fraction was used. [69] stated that for ecotoxicological assays, soil extracts (liquid fraction) could reflect the toxicity of the soil solid phase. However, metals can be present in the soil solution as free ion but also as complexes with organic or inorganic ligands. Therefore, partitioning of the metal ions between the solid phase and the solution depends on the composition of both phases [70]. An increase in organic carbon, for example, could lead to a decrease in metal availability [71], since organic carbon associated with clay minerals in the solid phase has a high capacity to bind metals and deplete the soil solution. However, the addition of vermicompost in our treatments did not reflect a better response for studied organisms, and only some improvements were obtained in the case of soil respiration (BV, GV, LV), for which the addition of vermicompost reflected a better response for this parameter. In this regard, some authors stated that organic carbon content influence soil respiration [72], while [73] verified that soils rich in organic carbon indeed had the highest CO_2_ efflux emitted, using same technique as that applied in this bioassay. In this study, it appeared that, in the solid fraction, the remediation techniques used recovered the polluted soil and presented little toxic effect. However, the correlation between the total and soluble forms of a pollutant was not always direct, and in this case, with natural polluted soils, we should consider the toxicity of multi-component chemical mixtures. For example, the solubility and toxicity of As are usually more tightly controlled by the soil properties than by the total amounts present in the sample [74,75]. In this sense, bioassays using the liquid phase extract can more readily reflect the behavior of mobile phases, evaluating the short-term risk of dispersion, solubilization and bioavailability of pollutants in the environment. In water fraction, according to the lettuce root reduction, all techniques were effective in the remediation, but if the results for algae or daphnid were observed, only GV, M and MV were effective techniques. According to soil properties, these three techniques were the remediation measures that showed a higher increase in pH. Organisms have their own sensitivity to pollutants and also show different responses under different exposure conditions [76]. It is generally considered that pH is one of the major factors controlling the toxicity of metals to aquatic organisms [77], and so, the observed results could be masked not only by the metal but also by pH effect.

Our results supported the theory that the use of only one bioassay may be useful for early identifying pollution problems, but it will not be sufficient as an indicator of toxicity pathways in order to select remediation methods under natural conditions. A metal that causes toxicity in some organisms may not do so in others, and the complexity of natural polluted soils due to the presence of mixture of elements, as well as different soil properties and constituents, can mask the effects attributed to the different metals involved [78]. Bioassays performed with samples taken from the polluted site after a period of application of the treatments allow assessing the real risk of pollution under natural conditions [79].

### 3.4. Effect of Soil Properties and Metal Availability in Organism Toxicity

Figure 3 is the result obtained after applying a non-metric multidimensional scaling (NMDS) (stress = 0.02), considering soil properties, metal studied forms and the response to the bioassays performed. The results showed that the application of the selected amendment treatments, after one year, induced great distances from the contaminated soil sample (CS), although they did not show similitude to the recovered soil (RS). The application of L (landfarming) and B (biopiles) were not useful remediation techniques for the recovery of the contaminated soil, since they followed similar distribution to the contaminated soil sample (CS). According to metal(loid)s concentration, both water-soluble and bioavailable forms decreased compared to the CS. However, these two remediation techniques showed the lowest pH. Soil properties are known to be essential factors when comparing toxicity of polluted soils [19,21]. Usually, soil pH is an important path for decreasing solubility of several heavy metals, by precipitation, adsorption or co-precipitation processes [80,81]. However, in real conditions, mixture of elements leads to synergistic or antagonistic effects [11], and the toxic effects of some elements can be masked by others. In the pH range of the L and B soils samples (4.6–5), it is expected that some elements, such as Zn, Cu or Pb, reach greater mobility [82]. It is well-known that soil properties are the main factors affecting metal speciation and bioavailability [83], thus being directly related to the response of living organisms in metal polluted soils. The lowest distances from RS appear in the case of MV, followed by M, GV, G, and LV, which, according to the results observed previously, showed a good recovery of soil properties, low levels of metal(loid)s studied and the best results from the ecotoxicological approach, mainly for treatments with marble sludge (M) and gypsum (G), although the ecotoxicological response was worse for the LV treatment (landfarming + vermicompost).

From NMDS distances, and aiming at upgrading the treated polluted soil to similar conditions of the nearby recovered soil (RS), we could approximate that the best remediation technique used was the addition of marble sludge with vermicompost (MV). Previous studies using similar soil amendments in the same area suggested the most appropriate doses of application for both marble and vermicompost [84,85], which were followed in this study (equivalent to 50 t ha^−1^). Thus, according to the results obtained, performing future studies or remediation programs based on marble sludge amendments on a larger scale following the specified dose are encouraged and could be feasible. This is based on the elevated production of this residual material, which, in the near future, could even become a serious environmental problem to be addressed [86]. In particular, MV technique increased the CO_3_Ca and OC content and raised the pH to ranges of 7.2. In addition, MV was effective in reducing some water-soluble and bioavailable forms of Cu, Zn and Cd (Table 2). Soil carbonates, in addition to their direct effect in increasing soil pH, may affect metal solubility and availability through their surface interactions, providing specific adsorption or precipitation reactions [87,88]. Metal toxicity is also influenced by metal binding state, the complexation usually being with organic ligands, from OC, a way to decrease toxicity [68]. Finally, and according to ecotoxicological responses, it must be highlighted that with the results showed for the five bioassays selected, MV showed toxic results close to those obtained for the recovered soil sample (RS) (Figure 2). Contrary to the findings of [12], who deduced that biopiles were an effective remediation action, mainly due to the increase in pH, our results suggest the opposite. This is explained by the fact that [12] performed these remediation techniques in laboratory with a short period of establishment. In contrast, our results demonstrated that the solubility, bioavailability and toxicity of metal(loid)s may vary over time due to the influence of soil properties and, presumably, the reduction in soluble concentrations leads to a decrease in the risk of toxicity to the ecosystem [11,74,89]. Consequently, when dealing with soil pollution by metal(loid)s from mining industry, and aiming to effectively assess soil toxicity risk and the success of soil remediation techniques, the determination of both metal availability and the toxicity for several organisms after an implementation period should be considered. In the case of this study, a temporal approach of one year was followed to observe the evolution of the soil toxicity after treatments implementation, which helped us to ensure the best remediation technique used. This approach could therefore be reliable for application in future studies aimed at a comprehensive assessment of toxicity in polluted soils by metal(loid)s.

## 4. Conclusions

Overall, the use of the different techniques selected for the remediation of a polluted natural area improved the main soil properties and led to significant changes in the mobility and availability of metal(loid)s one year after their application, thus being effective in reducing their toxicity. Of the different treatments used, the best technique for the remediation of soils affected by residual pollution and for reducing metal(loid)s toxicity was the addition of marble sludge with vermicompost. This technique was the one that most improved the main soil properties and was effective in reducing water-soluble and bioavailable forms of Cu, Zn and Cd. Additionally, and according to the ecotoxicological responses, this technique showed toxic results close to those obtained for the recovered soil.

The ecotoxicological approach performed to evaluate the responses of living organisms to metal(loid)s toxicity after remediation showed that their response to treatments differed depending on whether the solid or the aqueous fraction was used. According to the tests performed in soil-solid fraction, a positive response was found for all treatments compared to the contaminated soil, and so, the remediation techniques used seemed to effectively recover the polluted soil and presented little toxic effect. However, the results obtained from the toxicity tests performed in soil-water fraction showed high toxicity for some of the remediation techniques used. 

Our results highlighted that the use of a single bioassay may not be sufficient as an indicator of toxicity pathways to select soil remediation methods, and so, the joint determination of metal availability and ecotoxicological response of several organisms will be determinant for the correct establishment of any remediation technique carried out under natural conditions, where factors such as multi-stress or climatic conditions are involved.

## Figures and Tables

**Figure 1 toxics-11-00298-f001:**
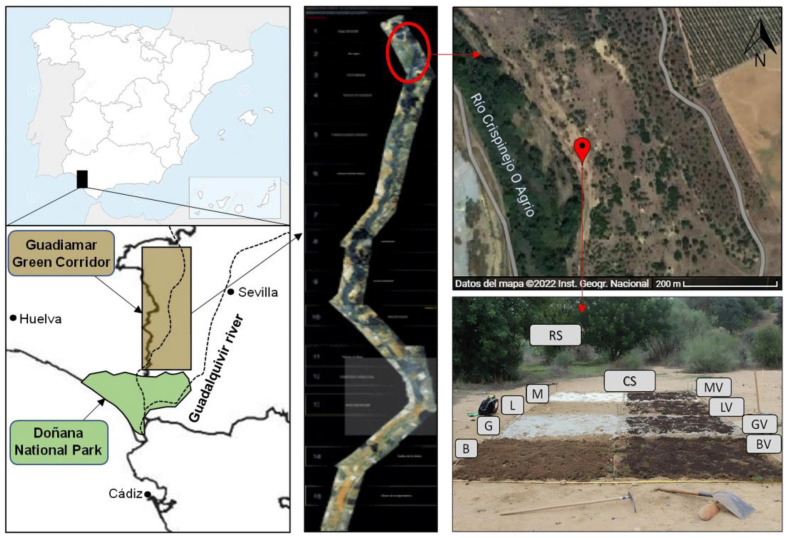
Study area, plot localization in the unvegetated soil nearby to the mine and treatments used for the remediation.

**Figure 2 toxics-11-00298-f002:**
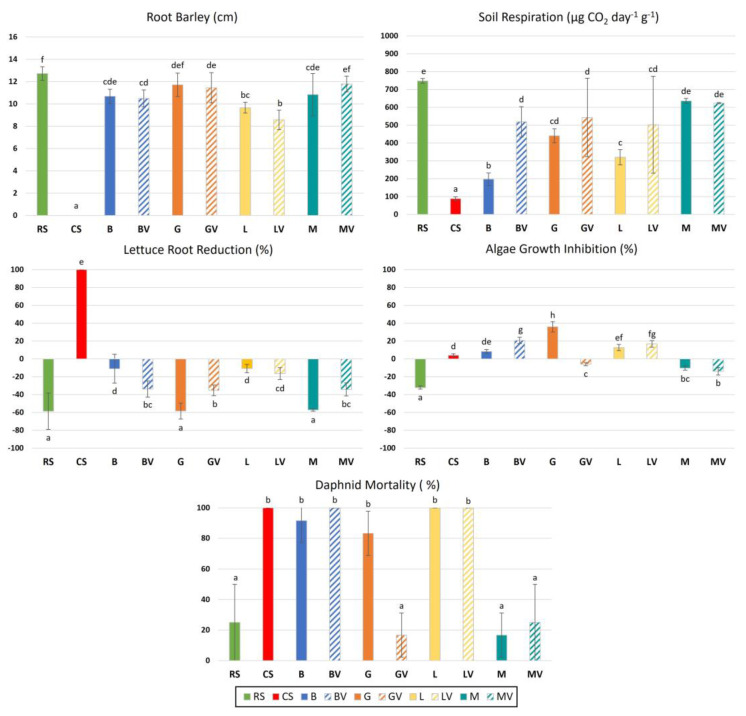
Ecotoxicological tests carried out in: a, soil-solid fraction: root barley elongation (cm) and soil induced respiration (µg CO_2_ day^−1^ g^−1^ soil) and b, in soil-water fraction: lettuce root reduction (%), algae inhibition growth (%) and daphnid mortality (%). Block bars represent the recovered soil (RS), the control polluted soil (CS); and the different studied amendments (B: Biopiles, G: Gypsum, L: Landfarming, M: Marble sludge). Striped bars indicate the use of composite amendments, with the addition of vermicompost (BV, GV, LV and MV). Lowercase letters indicate significant differences among treatments according to Duncan post hoc test (*p* < 0.05).

**Figure 3 toxics-11-00298-f003:**
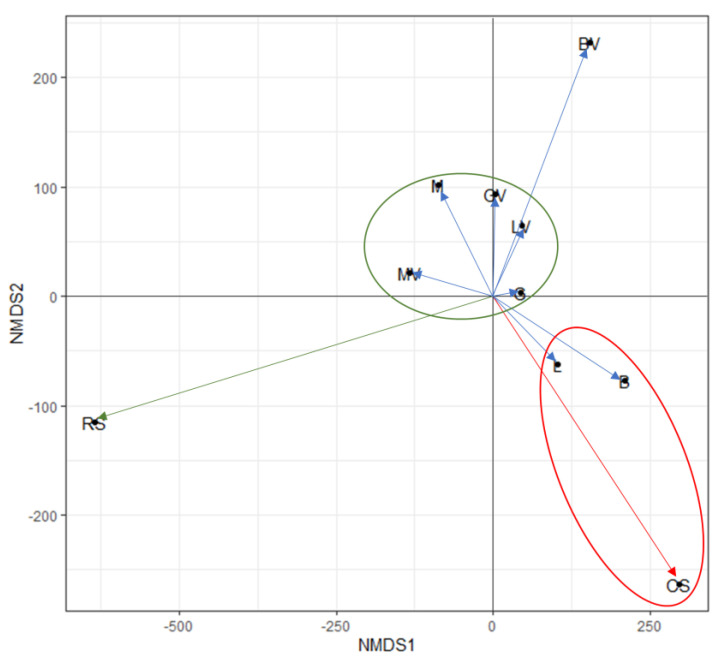
Non-metric multidimensional scaling (NMDS) for the different studied soils and treatments: the recovered soil (RS), the control polluted soil (CS), and the different studied amendments (B: Biopiles, G: Gypsum, L: Landfarming, M: Marble sludge; and the same with the addition of vermicompost (BV, GV, LV and MV)). The NMDS was carried out including the soil physicochemical properties; the total (Xx_T), bioavailable (Xx_B) and water-soluble (Xx_W) content of Cu, Zn, As, Pb and Cd; and the results for the five performed ecotoxicological tests.

**Table 1 toxics-11-00298-t001:** Description of the treatments used for soil remediation and the corresponding descriptor acronym.

Acronym	Treatment	Description
RS	RECOVERED SOIL	Soil naturally revegetated within the affected area
CS	CONTAMINATED SOIL	Unvegetated soil patches within the affected area
B	50% BIOPILES	50% *w*/*w* mixture of RS and CS
BV	BIOPILES + VERMICOMPOST	50% *w*/*w* mixture of RS and CS + vermicompost
G	GYPSUM	Gypsum mining spoil
GV	GYPSUM + VERMICOMPOST	Gypsum mining spoil + vermicompost
L	LANDFARMING	Soil crust breaking through tillage
LV	LANDFARMING + VERMICOMPOST	Soil crust breaking through tillage + vermicompost
M	MARBLE SLUDGE	Marble cutting and polishing residue
MV	MARBLE SLUDGE + VERMICOMPOST	Marble cutting and polishing residue + vermicompost

**Table 2 toxics-11-00298-t002:** Main soil properties for the different remediation techniques (see Table 1) one year after field application. EC: electrical conductivity; CO_3_Ca: calcium carbonate content; OC: organic carbon; N: total nitrogen; C/N: carbon-to-nitrogen ratio; CEC: cation exchange capacity; AWC: available water content.

Sample	pH	EC	CO_3_Ca	OC	N	C/N	Silt	Clay	CEC	AWC
		dS/m	%	%	%		%	%	cmol_c_/kg	%
RS	6.7 cd	1.0 a	2.30 a	1.28 e	0.11 d	11.4 c	28.1	15.3	11.4	15.1
CS	3.3 a	4.1 c	1.25 a	0.36 a	0.07 a	5.6 a	26.6	15.5	7.0	8.8
B	4.6 b	2.2 b	0.93 a	0.71 bc	0.09 bc	7.8 b	27.8	21.8	8.9	12.3
BV	5.1 b	2.2 b	1.07 a	0.90 cd	0.10 cd	8.6 b	32.7	23.0	10.5	12.4
G	6.0 c	2.3 b	1.25 a	0.61 ab	0.08 ab	8.0 b	27.4	22.2	8.6	11.7
GV	6.0 c	2.3 b	1.12 a	1.00 d	0.11 cd	9.0 b	27.3	19.6	9.4	11.9
L	5.0 b	2.2 b	1.05 a	0.49 ab	0.06 a	7.9 b	24.9	14.8	9.9	12.2
LV	4.3 b	2.2 b	0.99 a	0.72 bc	0.10 cd	7.5 ab	28.2	22.2	13.6	11.2
M	7.0 d	2.2 b	8.35 b	0.59 ab	0.06 a	9.4 bc	30.9	16.7	8.5	12.7
MV	7.2 d	2.3 b	3.18 a	0.99 d	0.11 d	8.8 b	29.5	20.7	9.9	14.4

Lowercase letters indicate significant differences among treatments according to Duncan post hoc test (*p* < 0.05).

**Table 3 toxics-11-00298-t003:** The total (Xx_T) soil concentration, the bioavailable concentration (Xx_B), and the water-soluble concentration (Xx_W) of the studied trace elements (Cu, Zn, As, Pb and Cd) for the recovered soil (RS), the contaminated soil (CS), and the different treatments used (see Table 1) one year after field application. Values in mg kg^−1^ dry soil.

Sample	*RS*	*CS*	B	BV	G	GV	L	LV	M	MV
Cu_T	248 e	149 abc	185 d	170 cd	142 ab	155 abc	160 bcd	151 abc	130 a	136 ab
Zn_T	926 a	359 b	316 b	335 b	235 b	232 b	319 b	213 b	240 b	349 b
As_T	209 a	309 bc	330 c	398 d	292 bc	326 bc	306 bc	317 bc	284 b	289 bc
Pb_T	314 a	430 b	560 c	838 d	496 bc	560 c	547 c	548 c	492 bc	436 b
Cd_T	4.4 bcd	6.6 e	5.0 cde	6.1 de	4.0 bc	2.9 ab	4.3 bcd	6.1 de	2.3 a	5.6 cde
Cu_B	67.96 d	37.95 c	33.59 bc	33.90 bc	31.52 bc	32.37 bc	29.96 bc	29.54 bc	16.87 a	24.45 ab
Zn_B	105.90 ab	239.84 b	78.91 a	72.35 a	42.30 a	46.70 a	96.81 ab	57.08 a	18.94 a	56.87 a
As_B	1.58 c	0.88 bc	0.11 ab	0.24 ab	0.13 ab	0.88 bc	<0.01	0.47 ab	0.09 ab	0.30 ab
Pb_B	2.47 b	0.30 ab	2.19 ab	1.65 ab	0.13 a	0.61 ab	<0.01	0.19 a	0.07 a	0.21 a
Cd_B	1.34 c	0.74 b	0.52 ab	0.53 ab	0.40 ab	0.37 a	0.46 ab	0.30 a	0.26 a	0.43 ab
Cu_W	0.41 a	19.30 b	1.09 a	0.40 a	0.15 a	0.22 a	0.65 a	0.95 a	0.15 a	0.18 a
Zn_W	0.56 a	206.14 b	31.95 a	18.38 a	0.22 a	0.22 a	38.76 a	38.16 a	<0.01	<0.01
As_W	0.25 b	0.10 ab	0.06 a	0.06 a	0.05 a	0.05 a	0.05 a	0.06 a	0.07 a	0.06 a
Pb_W	0.21 a	0.10 a	0.10 a	0.08 a	0.06 a	0.04 a	0.05 a	0.05 a	0.07 a	0.02 a
Cd_W	<0.01	0.78 b	0.18 a	0.08 a	0.01 a	0.01 a	0.16 a	0.14 a	<0.01	<0.01

Lowercase letters indicate significant differences among treatments according to Duncan post hoc test (*p* < 0.05).

## Data Availability

The data presented in this study are available on request from the corresponding author.

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
