# Peer review of "Ecotoxicological Assessment of Polluted Soils One Year after the Application of Different Soil Remediation Techniques"

_toxics, 2023, doi:10.3390/toxics11040298_

Round 1
Reviewer 1 Report
This work is of interest to readers of toxics and also other scientists. Please find further comments to improve the quality of this work below:
-line 40 – maybe specify the metal(loid)s. I stumbled over the concentration of uranium in some fertilizers recently (https://doi.org/10.1007/s11356-022-24574-5) that still keeps me puzzled…
-line 51: what material was mined?
-maybe also discuss how much of the treatment material would be needed and if this material would be available if you plan to upscale these experiments.
Reviewer 2 Report
This is an interesting piece of research on important environmental matters. Please see the following comments. Please in revision answer each comment on a separate doc document (not in the manuscript) stating also the lines in the amended manuscript that you did the corrections
1) some English language mistakes should be corrected. I have given only some examples. please proofread by a fluent English speaker and correct. Please use smaller and more concise sentences
eg replace
Selected remediation 13 treatments were left during one year in field exposed to real conditions, and subsequently five 14 ecotoxicological tests were carried out using different organisms and both the solid and aqueous 15 fraction of the amended soils.
with
selected remediation 13 treatments were applied in a field exposed to real conditions and they were evaluated one year after the application. More specifically five 14 ecotoxicological tests were carried out using different organisms on either the solid or the aqueous (leachate) 15 fraction of the amended soils.
replace
assessment and eventual remediation actions in relation to the PTEs 34 concentration
with
assessment and eventual remediation actions because of unacceptable PTEs 34 concentration
and some more. please proofread
2) what CO3Ca content means?
3) the introduction is quite interesting but very long. Please reduce to 2/3 of present length. Please transfer some of this information to discussion where appropriate
4) in the final sentences of the discussion please include a) in summary the problem b) what your research does c) what you aim to elucidate d) why this is important for an international audience
5) in materials and methods for all materials and apparatuses eg flasks, ICP-MS, reagents, centrifuge, sieves, TOC analyzer, pHmeter, Petri dished, all reagents give model of the apparatus if available, manufacturer, city and country of origin. For all methods described either give the protocol in detail or refer to a published protocol
6) I have no problem with the statistical analysis which looks sound. However what is a non-metric multidimensional scaling (NMDS and why for example no PCA was used? what it will explain?
7) Since you have many results I believe it is better for the reader to break into Results section and in Discussion sention because right now the text is too long for the reader. In the results section you should show only your Figures and Tables with a VERY short explanation in the text and in the discussion you should compare your results to bibliography without quoting actual values if possible
8) I cannot really understand Fig 2. First of all I do not understand what you mean by . Degraded bars indicate the use of composite amendments, 345 with the addition of vermicompost (BV, GV, LV and MV). what are degraded bars here? also I cannot see any statistical analysis on the results, why is this?
9) regarding the very important finding that total concentrations of metals are quite different that eg water soluble concentrations in relation to actual risk this should be highlited more in your study. Please also see and compare if possible with Giannakis I., Emmanouil C., Mitrakas M., Manakou V., Kungolos A.
Chemical and ecotoxicological assessment of sludge-based biosolids used for corn field fertilization
(2021) Environmental Science and Pollution Research, 28 (4), pp. 3797 - 3809,
